# miR-302 regulates pancreatic progenitor pool and pancreatic size

Ziyue Z. Yang[1,2], Caroline G. Snider[3] and Ronald J. Parchem[1,*]

## ABSTRACT

Disruptions in pancreatic development can lead to health issues such as pancreatic agenesis and congenital diabetes mellitus. Understanding pancreatic organogenesis is critical for elucidating disease mechanisms and developing regenerative therapies. The pancreas consists of endocrine and exocrine cells, both of which are derived from multipotent progenitor cells (MPCs). MPC proliferation and differentiation are tightly controlled by multiple mechanisms, including post-transcriptional regulation by miRNAs. However, these regulatory factors are not fully understood. Here, we profiled miRNA expression in MPCs and identified that *mir-302* was highly enriched during the earliest stages of pancreatic development. Loss of *mir-302* resulted in reduced pancreatic size without altering the proportions of endocrine and exocrine cells at E17.5, suggesting that *mir-302* regulates the number of MPCs rather than their differentiation. Transcriptomic analysis at E10.5 revealed that *mir-302* modulates genes associated with the Wnt signaling pathway and cell cycle progression. Notably, loss of *mir-302* prolonged the S phase in MPCs, resulting in slower cell proliferation and a smaller MPC pool at E10.5. These findings provide the first comprehensive miRNA profile during early pancreatic development and establish *mir-302* as a critical regulator of MPC number and pancreas size.

KEY WORDS: miRNA, Pancreatic development, Progenitor cells, miR-302

## INTRODUCTION

The pancreas is composed of endocrine and exocrine cells and has diverse and vital roles in nutrition and metabolism. Endocrine cells secrete hormones essential for regulating blood glucose and maintaining energy homeostasis, whereas exocrine cells synthesize and secrete digestive enzymes into the duodenum to facilitate digestion. Abnormalities during pancreatic development can result in pancreatic diseases, such as diabetes mellitus, pancreatitis, and pancreatic agenesis (Holemans et al., 2003; Lorberbaum et al., 2020; Yang and Parchem, 2023). Understanding pancreatic development is indispensable for developing novel therapeutic approaches for pancreatic diseases, as mechanisms underlying pancreatic

regeneration often overlap with those controlling pancreatic development. Regeneration of acinar cells is necessary for resolving inflammation associated with pancreatitis, and impaired exocrine regeneration can contribute to the development of lethal pancreatic cancer (Stanger and Hebrok, 2013; Zhou and Melton, 2018). Increasing functional β-cell mass presents as the optimal therapeutic strategy for diabetes (Benthuysen et al., 2016), either by promoting endogenous β-cell regeneration and proliferation or by transplanting exogenous β-cells (Ramzy et al., 2021; Shapiro et al., 2021).

Pancreatic agenesis is a rare congenital defect caused by genetic mutations (Agenesis of the dorsal pancreas - About the Disease - Genetic and Rare Diseases Information Center; Allen et al., 2011; Xuan et al., 2012). The size of the mature pancreas is largely determined by the number of progenitor cells early in the development (Stanger et al., 2007). Multipotent progenitor cells (MPCs) give rise to both endocrine and exocrine pancreatic cell lineages (Jørgensen et al., 2007; Shih et al., 2013). Unlike the liver, which can compensate for progenitor cell loss and achieve normal size, the pancreas size remains smaller if progenitor cell number is compromised early during development (Stanger et al., 2007). MPC proliferation and differentiation are tightly controlled processes. During the primary transition (E8.5 to E10.5 in mice), MPCs are initially induced and specified in the posterior foregut endoderm and then proliferate to populate the pancreatic progenitor pool while remaining undifferentiated. MPCs are tightly packed within the pancreatic bud and begin branching into the tip and trunk domains around E10.5. Subsequently, MPCs adopt either tip or trunk fate cell fates, which are programmed for different pancreatic cell identities. This process is referred to as the secondary transition (Jørgensen et al., 2007; Shih et al., 2013). Several transcription factors have been shown to control MPC number and pancreatic size, including Gata4, Gata6 (Carrasco et al., 2012; Shi et al., 2017; Xuan et al., 2012), Ptf1a (Krapp et al., 1998), Pdx1 (Offield et al., 1996), Brg1 (Spaeth et al., 2019) and Sox9 (Seymour et al., 2007). Understanding the mechanisms regulating MPC proliferation and differentiation will offer valuable insights into developing therapeutic approaches for diverse pancreatic disorders.

Post-transcriptional regulation by microRNAs (miRNAs) is also critical for proper pancreatic development and function (Dumortier and Van Obberghen, 2012; Fukasawa et al., 2006; Lynn et al., 2007; Prévot et al., 2013). miRNAs are small non-coding RNAs that mediate post-transcriptional regulation by binding to the 3′ untranslated regions (3′UTRs) of target mRNAs, thereby silencing gene expression. The miRNA regulatory pathway plays an important role in pancreatic development. Inactivation of Dicer1, a key enzyme in miRNA biogenesis, in MPCs leads to pancreatic agenesis and differentiation defects across all pancreatic lineages (Lynn et al., 2007; Prévot et al., 2013). In addition, Dicer1 deletion in endocrine lineages using Ngn3-Cre results in the upregulation of neuronal genes, suggesting that miRNAs are essential for repressing neuronal gene expression in the pancreas (Kanji et al., 2013). Ago2, another key enzyme in miRNA biogenesis and function, mediates compensatory β-cell expansion during insulin resistance (Tattikota et al., 2014). Individual miRNAs have also been studied in the

[1]Center for Cell and Gene Therapy, Stem Cells and Regenerative Medicine Center, Department of Neuroscience, Department of Molecular and Cellular Biology, Translational Biology and Molecular Medicine Program, Baylor College of Medicine, One Baylor Plaza, Houston, TX 77030, USA. [2]Development, Disease Models and Therapeutics Graduate Department, Baylor College of Medicine, One Baylor Plaza, Houston, TX 77030, USA. [3]Department of BioSciences, Rice University, Houston, TX 77005, USA.

*Author for correspondence (ronald.parchem@bcm.edu)

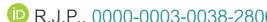 R.J.P., 0000-0003-0038-2806

context of pancreatic development (Domínguez-Bendala et al., 2023; Dumortier and Van Obberghen, 2012; Yang and Parchem, 2023). For example, *mir-375* deficiency during pancreatic development leads to decreased β-cell mass and hyperglycemia (Poy et al., 2009).

To elucidate the role of miRNAs in MPC proliferation and differentiation, we profiled miRNA expression during the primary transition of pancreatic development. Our analysis revealed that *mir-302* was enriched at this stage. The miR-302 family is highly conserved across vertebrates (Chen et al., 2015). It is highly expressed in pluripotent stem cells and promotes stemness while inhibiting differentiation (Chen et al., 2015; Gao et al., 2015; Lin et al., 2011; Parchem et al., 2014). Previously, *mir-302* has been shown to regulate proliferation and differentiation of neural crest cells (Keuls et al., 2020, 2023; Parchem et al., 2015), which are multipotent stem cells located at the neural plate border that migrate throughout the embryo and differentiate into multiple cell types (Mayor and Theveneau, 2013). However, the role of *mir-302* in pancreatic progenitors remains unexplored. Using genetic *mir-302* reporter and knockout mouse models, we discovered that *mir-302* regulates MPC number, thereby influencing pancreatic size later in development. These findings establish *mir-302* as a regulator of the MPC pool number and the ultimate size of the pancreas.

## RESULTS

### miRNA expression profile during the primary transition

Previous studies have characterized miRNA expression profile in the pancreas during the secondary transition, when progenitor cells differentiate into specific pancreatic cell lineages (Baroukh et al., 2007; Joglekar et al., 2007; Lynn et al., 2007). To obtain a comprehensive understanding of miRNAs involved in primary transition, we employed lineage tracing and isolated pancreatic progenitors at multiple time points. We used *Pdx1-Cre* (Gu et al., 2002) to label MPCs from embryonic day (E) 9.5 to E11.5, and *Foxa2^mcm^ (Foxa2^CreER^)* (Park et al., 2008) to label the endoderm at E8.5 (Fig. 1A,B). We used fluorescence-activated cell sorting (FACS) to isolate *Foxa2^mcm^*-labeled cells from the whole embryos, and *Pdx1-Cre*-labelled cells from the trunk region (Fig. 1B). Cells were sorted into Trizol LS for small RNA sequencing (Keuls and Parchem, 2020) (Tables S1-S2). Principal component analysis (PCA) revealed distinct clustering between *Foxa2* and *Pdx1* lineages along principal component 1 (PC1). Among the *Pdx1-Cre*-labeled samples, E10.5 and E11.5 clustered closely together, separating from E9.5 along PC2 (Fig. 1C).

We employed K-means clustering to categorize miRNAs based on their expression patterns in pancreatic progenitors (*Pdx1-Cre*-labeled) (Table S3). This analysis identified three major clusters of miRNAs during early pancreatic development from E9.5 to E11.5, representing the dominant patters of: (1) upregulated, (2) downregulated, and (3) dynamically expressed (Fig. 1D, Table S4). To identify the miRNAs that best represented each cluster, we ranked miRNAs by their Euclidean distance to the cluster centroid in the scaled expression space (see the Materials and Methods). The top-ranked miRNAs (nearest to the centroid) exhibited the most prototypical expression patterns of their respective clusters and were selected as representative miRNAs for each cluster. Cluster 1 was enriched with miRNAs associated with differentiation and developmental processes. For example, *mir-106b* has been implicated in spermatogonial differentiation (Tong et al., 2012), and *mir-100* and *mir-99b* have been shown to regulate hematopoietic stem and progenitor cell proliferation (Emmrich et al., 2014; Gerrits et al., 2012). In contrast, cluster 2 included pluripotency-associated miRNAs that were downregulated

during this window, such as *mir-290*, *mir-302* (Gu et al., 2016), and *mir-200* (Balzano et al., 2018; Wang et al., 2013a). *mir-375* is known to regulate β-cell proliferation and function (Eliasson, 2017; El Ouaamari et al., 2008; Poy et al., 2009) and its expression has been detected in the pancreas at E14.5 (Lynn et al., 2007). Interestingly, we found that it was a representative member in cluster 3, with expression peaking at E10.5 during the primary transition. Cluster 3 also included many novel miRNAs, such as *mir-370*, *mir-127*, *mir-434*, and *mir-200b*, that may be key regulators of early pancreatic development and require further characterization.

To examine miRNA expression dynamics in greater detail, we compared the expression levels between adjacent time points. Between E8.5 and E9.5, when the earliest pancreatic progenitors are induced, *let-7* family, which is known to promote differentiation, was upregulated (Büssing et al., 2008; Reinhart et al., 2000; Zhao et al., 2010). In contrast, the stem cell-associated miRNAs, *mir-302* and *mir-290* families (Barroso-DelJesus et al., 2008; Gao et al., 2015; Gu et al., 2016; Parchem et al., 2014), were downregulated (Fig. 1E). Interestingly, *mir-122*, the most abundant miRNA in the liver (Kim et al., 2011; Lagos-Quintana et al., 2002), was also detected in the earliest pancreatic progenitors. This observation aligns with the fact that the ventral pancreas and liver are specified concurrently within the same endodermal domain (Deutsch et al., 2001; Nowotschin et al., 2019; Zaret and Grompe, 2008). *mir-7* family members play essential roles in β-cell differentiation and function (Latreille et al., 2014; Nieto et al., 2012; Wang et al., 2013b), and are expressed during primary transition (Nieto et al., 2012). We found that *mir-7* family members were significantly upregulated between E8.5 and E9.5 as well (Fig. 1E). From E9.5 to E10.5, the expression of *mir-290* and *mir-302* family members decreased as differentiation progresses. Interestingly, *let-7* family members also showed decreased expression, suggesting that they may have stage-specific functions in regulating pancreatic development. *mir-375*, a well-characterized miRNA during pancreatic development (Eliasson, 2017; Joglekar et al., 2009; Poy et al., 2004, 2009), was notably upregulated at E10.5, which was in line with its peak expression at this stage (Fig. 1D,F). Consistent with PCA clustering (Fig. 1C), only a few miRNAs were differentially expressed between E10.5 and E11.5, during which *mir-290* and *mir-302* family members were further downregulated (Fig. 1G).

In addition to miRNAs known to regulate pancreatic development, we identified several differentially expressed miRNAs between developmental stages that have been implicated in pancreatic diseases. For example, *mir-181* and *mir-183* promote KRAS-driven tumorigenesis in pancreas and lung (Miao et al., 2016; Valencia et al., 2020; Yang et al., 2019), whereas *mir-143* targets *Kras* and suppresses pancreatic ductal adenocarcinoma progression (Hu et al., 2012; Xie et al., 2019). Moreover, *mir-24* has also been found to inhibit β-cell proliferation and insulin secretion by targeting maturity-onset diabetes of the young (MODY) genes, *Hnf1α* and *Neurod1* (Zhu et al., 2013). Elucidating the functions of these miRNAs during normal development will enhance our understanding of their roles in pathological conditions.

To better understand the predominant miRNAs at each developmental stage, we analyzed miRNA expression based on their seed sequences. Raw small RNA sequencing read counts from mature miRNAs sharing identical seed sequences (TargetScan Mouse 8.0) (McGeary et al., 2019) were grouped together (Table S5) and normalized using the DEseq2 package (Love et al., 2014) (Table S6). Notably, more than 70% of the total reads were attributed to the top twenty enriched seed sequences at each time point (Fig. 1H). Interestingly, over half of these top 20 seed sequences were shared

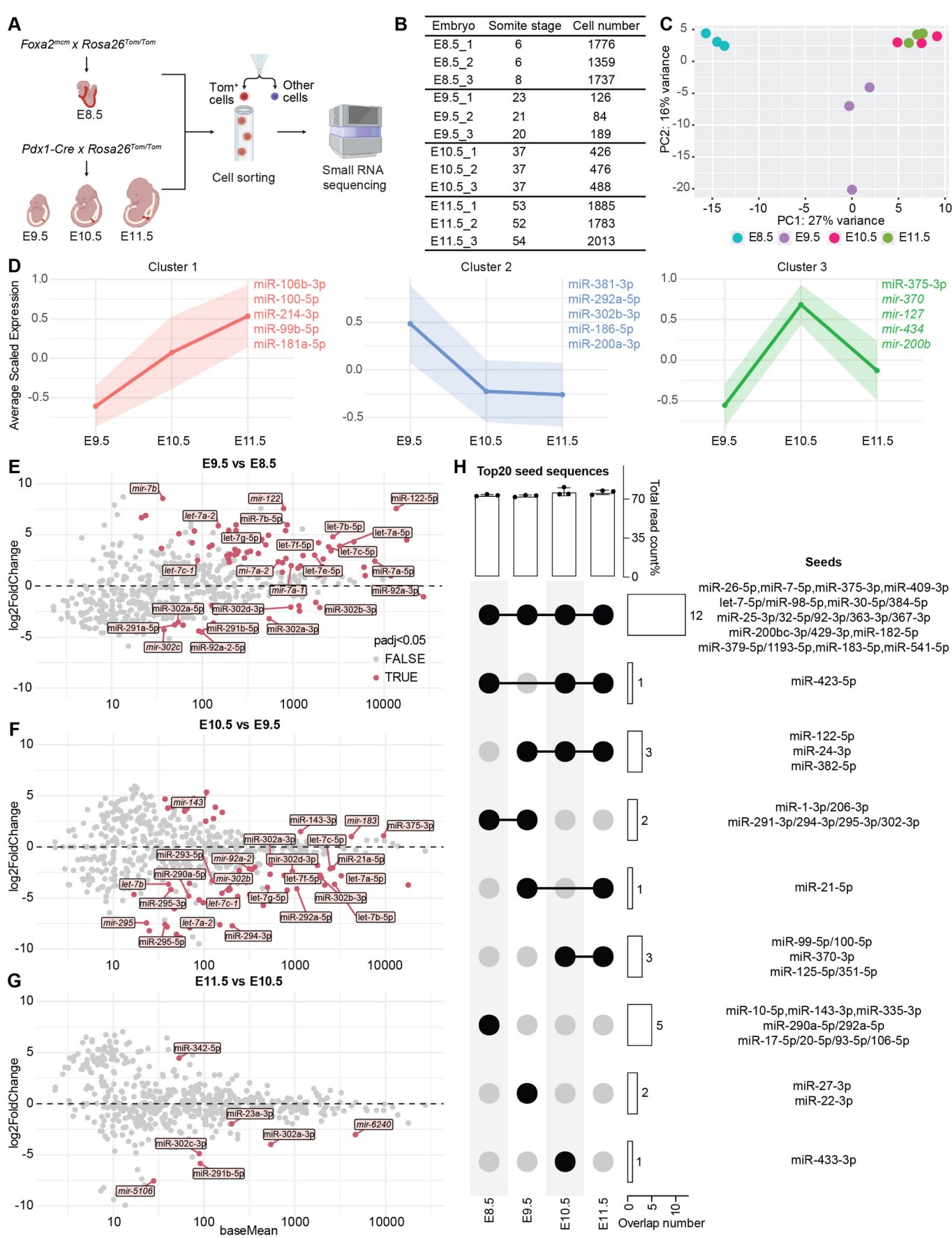

Fig. 1. See next page for legend.

**Fig. 1. miRNA expression profile during early pancreatic development.**
(A) Schematic of lineage tracing and cell sorting strategy to isolate
endoderm (*Foxa2^mcm*, E8.5) and pancreatic progenitors (*Pdx1-Cre*, E9.5 to
E11.5) from E8.5 to E11.5. Tom, tdTomato. Created in BioRender by
Yang, Z. (2025). https://BioRender.com/tc5eybj. This figure was sublicensed
under CC-BY 4.0 terms. (B) Summary of embryos used in A. (C) PCA plot
showing sample clustering based on miRNA expression profile from E8.5 to
E11.5. (D) K-means clustering of miRNAs based on expression pattern from
E9.5 to E11.5. Normalized read counts were obtained using DESeq2 and
scaled with Z-score normalization. Solid lines represent average scaled
expression (mean) of all miRNAs in each cluster at each time point and the
shaded area indicates the standard deviation. Only miRNAs with normalized
counts >100 in at least three samples were included in the clustering. The
most representative miRNAs of each cluster were labeled in corresponding
colors. (E-G) MA plot of differentially expressed miRNAs between time
points. Colored dots are significantly changed miRNAs with a *P*adj<0.05,
while grey dots are not significantly changed. Log2Foldchange and
baseMean were obtained using DESeq2. (H) UpSet plot showing the
overlap of the top 20 enriched miRNA seed sequences across
developmental stages. The top bar graph represents the proportion of total
reads accounted for by the top 20 seeds at each stage. The right bar chart
indicates the number of seeds shared among different stages.

across all four stages, suggesting that these miRNAs may play
important roles during the primary transition. In fact, *mir-7* (Kredo-
Russo et al., 2012; Latreille et al., 2014; Nieto et al., 2012; Wang et al.,
2013b), *mir-375* (El Ouaamari et al., 2008; Kloosterman et al., 2007;
Lahmy et al., 2014; Poy et al., 2004; Poy et al., 2009; Tattikota et al.,
2013; Tattikota et al., 2014), and *mir-30* (Joglekar et al., 2009; Kim
et al., 2013; Liao et al., 2013; Tang et al., 2009; Zhao et al., 2012) were
among the top 20 seeds across all four time points and have been
characterized in diabetes (Fig. 1H). *mir-541* was previously reported
to be expressed in the developing pancreas at E14.5 (Lynn et al.,
2007), and we found it was also enriched during the primary transition
(Fig. 1H). These results validated that our profiling approach was able
to reveal pancreatic miRNAs. Additionally, we identified many novel
miRNA families enriched during the primary transition that could be
highly relevant, such as *mir-290*, *mir-302*, and *mir-423*, which have
not been implicated in pancreatic function or diseases. Some families
exhibited stage-specific enrichment. For example, the *mir-290* and
*mir-302* families were enriched at E8.5 and E9.5. In summary, our
profiling of miRNA expression during early pancreatic development
provides valuable insights into pancreatic development and may
inform new strategies for regeneration medicine.

### *mir-302* is expressed during the primary transition
Among the miRNAs identified during the primary transition, we
asked which miRNA could regulate MPC specification and
proliferation. Previous studies have demonstrated that *mir-302*
regulates proliferation and differentiation of neural crest cells (Keuls
et al., 2020, 2023; Parchem et al., 2015), a population of multipotent
stem cells capable of giving rise to multiple cell types, similar
to MPCs in the pancreas. During the differentiation of human
embryonic stem cells into pancreatic lineages, miR-302 is highly
expressed during the formation of primitive gut tube and posterior
foregut (Fogel et al., 2015). Additionally, miR-302 is upregulated
in de-differentiated human pancreatic islet cells (Sebastiani et al.,
2018). Given that *mir-302* is highly enriched at E8.5 and E9.5,
when the earliest pancreatic progenitors are specified and start
to populate the progenitor pool, we hypothesized that *mir-302*
regulates MPC proliferation and differentiation. Small RNA
sequencing data showed that *mir-302* read counts were decreased
from E8.5 to E11.5 (Fig. 2A,B). To visualize *mir-302* expression
within pancreatic lineages, we employed a knock-in *mir-302-eGFP*
reporter (Parchem et al., 2015) and simultaneously used *Pdx1-Cre*

driving *Rosa26^LSL-tdTomato* (Madisen et al., 2010) to label pancreatic
progenitors (Fig. 2C). Consistent with the sequencing data, *mir-302*
was highly abundant at E9.5, decreased at E10.5 and E11.5, and was
minimally detected at E13.5 (Fig. 2D,E).

### Loss of *mir-302* reduces pancreatic size without altering cellular composition at E17.5
To investigate the role of *mir-302* during pancreatic development,
we utilized a *mir-302* genetic knockout (KO) mouse model.
Heterozygous *mir-302^+/−* females and males were crossed to
generate wild-type (WT) and KO embryos. Since *mir-302* KO
embryos are not viable beyond E18.5 (Parchem et al., 2015), we
dissected embryos at E17.5, the latest time point at which mature
and differentiated pancreatic tissue can be analyzed. Compared
with WT pancreas, the size of *mir-302* KO pancreas exhibited a
34.5% reduction at E17.5 (Fig. 3A-C). To assess cell proliferation,
we used bromodeoxyuridine (BrdU)-labeling for 30 min prior to
dissection (Spaeth et al., 2019) and compared the percentage of
BrdU-positive cells in the pancreas. There was no significant
difference in the proportion of cells in the S phase between KO and
WT in the pancreas at E17.5 (Fig. 3D,E). Suggesting that any
proliferation defects likely occurred at earlier developmental
stages.

Since *mir-302* was highly expressed in MPCs (Fig. 2), which give
rise to all pancreatic lineages, we next investigated whether *mir-302*
deficiency altered the proportions of differentiated pancreatic cell
types. When comparing the percentage of INSULIN, GLUCAGON,
and MUCIN1-positive cells in the pancreas, loss of *mir-302* did not
affect their abundance in the pancreas (Fig. 3F-I). The spatial
localization of these cells remained unchanged as well: INSULIN^+
β-cells were still centrally located within the islets, surrounded
by GLUCAGON^+ α-cells (Fig. 3F). Similarly, MUCIN1^+ lumens
were maintained at the apical center of the acini and surrounded by
E-CAD^+ epithelium cells (Fig. 3H). Altogether, these data indicate
that miR-302 regulates pancreatic size without altering cellular
composition.

### *mir-302* controls pancreatic size by controlling progenitor cell number
A previous study established that pancreatic size is determined by
the number of progenitor cells (Stanger et al., 2007). Given that *mir-
302* was enriched in early progenitors and has been shown to
regulate cell cycle and proliferation (Gao et al., 2015; Keuls et al.,
2020; Parchem et al., 2015; Tian et al., 2015), we hypothesized that
*mir-302* regulates pancreatic size by modulating progenitor cell
number. To test this, we analyzed the number of pancreatic progenitor
cells at E10.5 by quantifying the PDX1 positive pancreatic area. We
found that loss of *mir-302* led to a significant decrease in pancreatic
size by 31.2% at E10.5 (Fig. 4A,B), comparable to the size reduction
observed at E17.5 (Fig. 3C). These data are in line with the fact that
pancreatic size is established early in development and is not
compensated at later stages.

To understand how loss of *mir-302* reduced progenitor pool, we
sorted Pdx1^+ lineages from the trunk region in WT and KO embryos
(Fig. 4C,D) and compared their gene expression profiles (Table S7).
Consistent with the canonical role of miRNAs in downregulating
mRNA expression (Bartel, 2004), we observed that the majority of
the differentially expressed genes (DEGs) were upregulated upon
*mir-302* deficiency (606 upregulated versus 93 downregulated)
(Fig. 4E). Notably, almost half of these DEGs (333 of 699 DEGs)
were predicted *mir-302* targets [TargetScan Mouse 8.0 (McGeary
et al., 2019), Fig. 4E] (McGeary et al., 2019). Gene Ontology (GO)

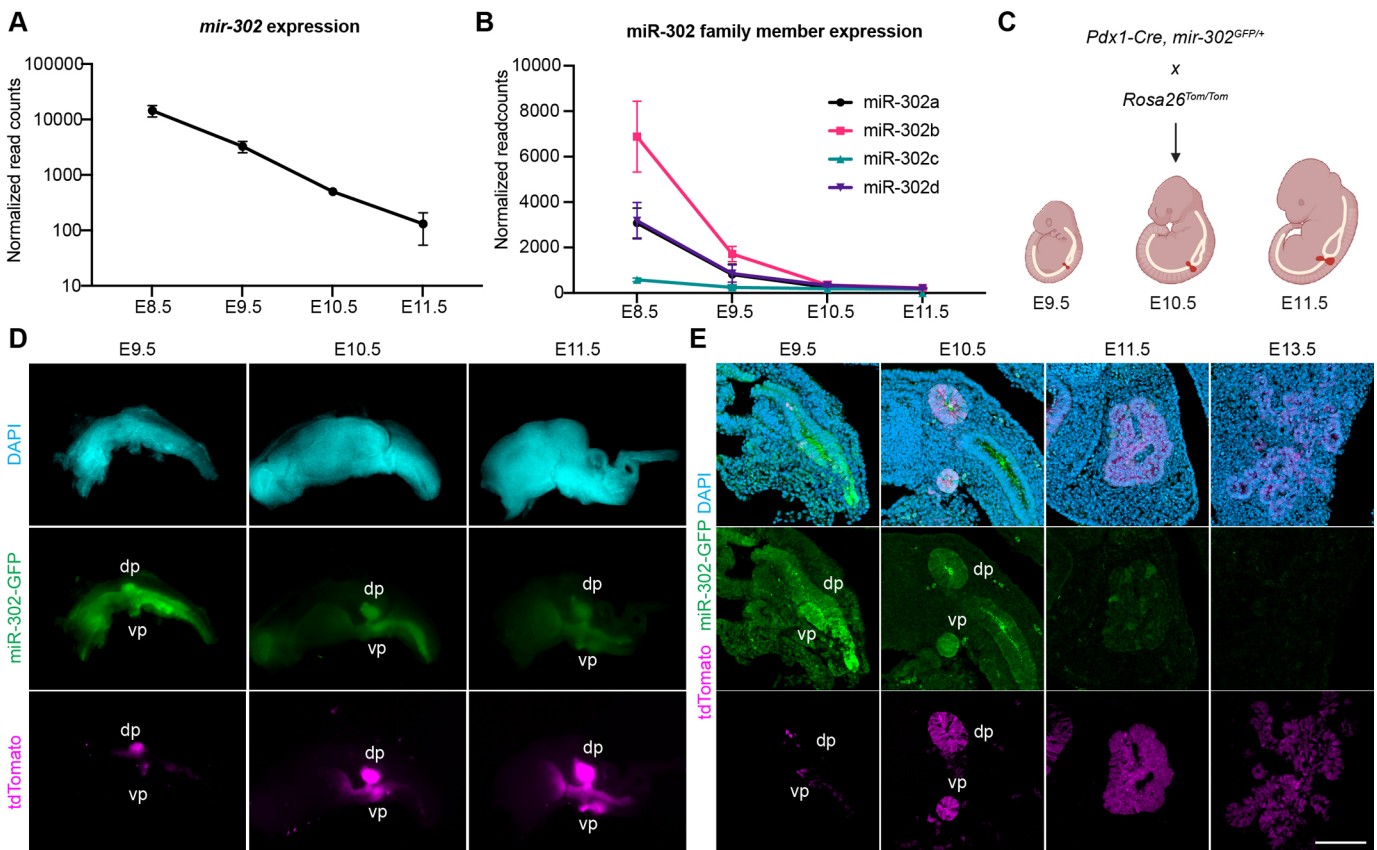

**Fig. 2. *mir-302* is downregulated during primary transition.** (A,B) Normalized read count of (A) total *mir-302* and (B) individual *mir-302* members across E8.5 to E11.5 in small RNA sequencing. (C) Schematic of lineage tracing strategy to label pancreatic cells in *mir-302* reporter mice. Tom, tdTomato. Created in BioRender by Yang, Z., 2025. https://BioRender.com/ouqmtr8. This figure was sublicensed under CC-BY 4.0 terms. (D) Whole-mount immunofluorescent images of dissected gut tube in C. Gut tubes from E9.5 to E11.5 were dissected and stained for DAPI (cyan), PDX1 (magenta), GFP (green). dp, dorsal pancreas; vp, ventral pancreas. (E) Sagittal sections showing the expression levels of *mir-302* in developing pancreas. Scale bar: 100 μm.

analysis showed that the upregulated DEGs were enriched in terms related to the Wnt signaling pathway, cell differentiation, and mRNA stability (Fig. 4F). The Wnt signaling pathway is known to promote cell cycle progression (Habib and Acebrón, 2022; Niehrs and Acebron, 2012; Tetsu and McCormick, 1999), and there were multiple predicted *mir-302* targets within this pathway that were upregulated in the absence of *mir-302* (Napolitano et al., 2023). For example, Lrp6 and Ccny regulate the G2/M transition (Davidson et al., 2009), Amfr and Fzd7 promote the G0/G1 progression (Asad et al., 2014; Liu et al., 2016; Wang et al., 2015), and Grk6, Sox2, Wnk1, Sulf2 and Rac1 facilitate the G1/S transition (Liu et al., 2014; Ren et al., 2016; Xu et al., 2017; Yoshida et al., 2010; Zhang et al., 2018; Zhu et al., 2016) (Fig. 4G). These findings suggest that miR-302 modulates progenitor cell number by regulating cell cycle progression.

To compare the number of cells in the S phase between KO and WT pancreas, we analyzed BrdU incorporation in MPCs at E10.5 after a 30-min pulse (Fig. 4A). Despite a reduction in the total number of progenitor cells in KO embryos (Fig. 4B), a higher proportion of these cells were in the S phase (Fig. 4H). In addition, the percentage of progenitors in the M phase was not different between WT and KO pancreas (Fig. 4I-J). There was no significant changes in apoptosis either (Fig. 4I,K). These findings indicate that *mir-302* deficiency prolonged the S-phase, resulting in a slower cell cycle, decreased proliferation, and a smaller progenitor cell pool.

## DISCUSSION

Pancreatic development involves multiple, precisely regulated mechanisms to ensure proper organogenesis. The specification and proliferation of MPCs are the earliest steps of pancreatic development, and these progenitors subsequently proliferate and differentiate into various pancreatic cell types. Defects in MPC proliferation and differentiation can cause diverse pancreatic diseases, which may potentially be treated by regenerative medicine approaches (Ellis et al., 2017). Our study specifically addressed the role of miRNAs during early pancreatic development by identifying miRNAs enriched in MPCs and revealing that *mir-302* regulates MPC proliferation and pancreatic size.

It is technically challenging to isolate and profile pancreatic tissue during early development due to insufficient material and potential contamination of non-pancreatic cells. We combined lineage tracing and cell sorting to ensure the accurate cell origin. We also leveraged two different Cre drivers, *Foxa2^mcm* and *Pdx1-Cre*, to label endoderm and pancreatic progenitors, which allowed us to achieve the first comprehensive profile of miRNA expression at early developmental stages. Our results identified some miRNAs known to regulate pancreatic development and function, as well as novel regulators such as *mir-302*. Interestingly, miR-302 is well-characterized in embryonic stem cells, where it maintains pluripotency and self-renewal by targeting cell cycle regulators and epigenetic modifiers (Chen et al., 2015; Keuls et al., 2020, 2023; Lee et al., 2013; Lin et al., 2011; Lipchina et al., 2011; Li et al., 2016). Its expression during early

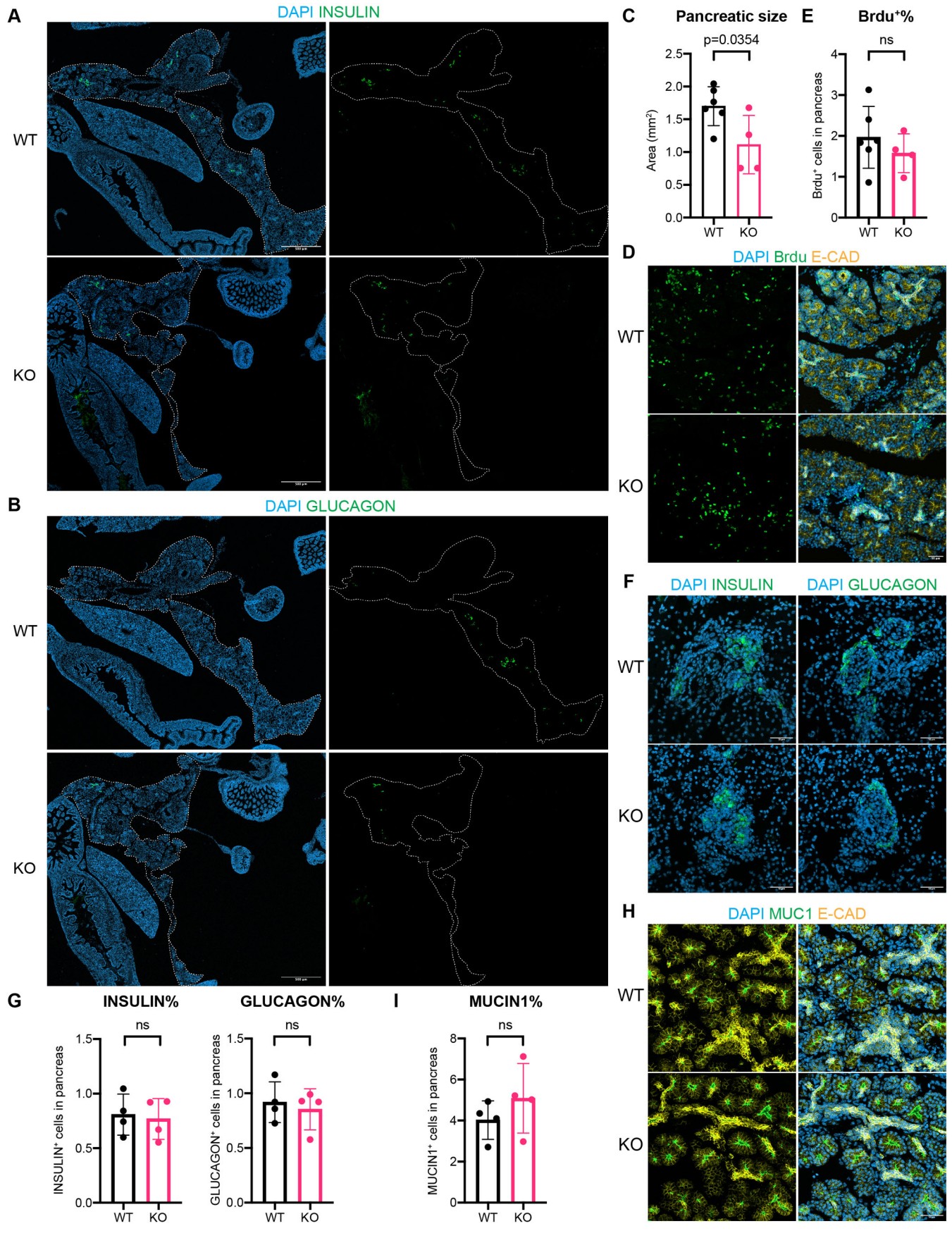

**Fig. 3.** See next page for legend.

Biology Open

**Fig. 3. *mir-302* regulates pancreatic size rather than cellular composition.** (A,B) Immunofluorescence (IF) staining of (A) INSULIN and (B) GLUCAGON in WT and KO pancreas at E17.5. The liver, pancreas, spleen, and intestine were dissected together from the embryos, and embedded and sectioned together coronally. The pancreas was outlined in white dashed line. Scale bars: 500 µm. (C) Quantification of pancreatic size outlined in dashed line in A and B. (D) IF staining of BrdU (green) and ECAD (yellow) after 30-min pulse before dissection at E17.5. Scale bar: 50 µm. (E) Quantification of BrdU+ cells in the pancreas in D. (F) Zoomed in images of INSULIN and GLUCAGON staining showing their localization in islets. Scale bars: 50 µm. (G) Quantification of INSULIN+ and GLUCAGON+ cells in the pancreas in A and B. (H) IF staining of MUCIN1 (green) and E-CADHERIN (yellow) showing their localization in the acini at E17.5. Scale bar: 50 µm. (I) Quantification of MUCIN1+ in the pancreas in H. *P*-value in (C,E,G,I) by Student's *t*-test. Each data point represented an embryo quantified with at least 12 coronal sections spaced equally across the pancreas.

pancreas development, as we observed, highlights potential parallels between pluripotent stem cell regulation and organ-specific progenitor regulation. Notably, while *mir-302* deficiency in neural crest cells promotes cell cycle progression (Keuls et al., 2020; Parchem et al., 2015), we found that in pancreatic progenitors, loss of *mir-302* leads to prolonged S phase and reduced proliferation. This suggests that miR-302's targets and regulatory effects are highly context-dependent, varying across tissues and developmental programs.

Consistent with the canonical function of miRNA in suppressing gene expression, we discovered more upregulated DEGs than downregulated ones. Apart from genes involved in the Wnt signaling pathway, we also found upregulation of *Sonic Hedgehog* (*Shh*) (Fig. 4G). In mouse mid-gestational embryos, *Shh* is expressed in the epithelium throughout digestive tract but inhibited by the notochord and excluded from the pancreatic region. In fact, elevated Shh levels impair pancreas formation, while inhibiting Shh within the gut leads to the formation of a heterotopic pancreas (Hebrok, 2003; Hebrok et al., 1998, 2000; Kim and Melton, 1998). Therefore, the upregulation of *Shh* upon *mir-302* loss could further inhibit the specification of pancreatic progenitors, particularly during E8.5 and E9.5 when *mir-302* expression peaks (Fig. 2).

Pancreatic size is limited by the number of progenitors (Stanger et al., 2007), and previous studies have identified DNA-binding transcription factors as the primary regulators of the pancreatic progenitor pool size, whose dysfunction leads to pancreatic agenesis (Carrasco et al., 2012; Krapp et al., 1998; Offield et al., 1996; Seymour et al., 2007; Shi et al., 2017; Spaeth et al., 2019; Xuan et al., 2012). Here, we reported an miRNA, *mir-302*, that regulates progenitor cell proliferation and pancreatic organ size. It remains to be determined whether other miRNAs compensate for *mir-302* loss, or whether parallel regulatory pathways buffer its effects on downstream differentiation programs. Exploring these compensatory mechanisms could reveal broader insights into progenitor pool homeostasis.

Moreover, our findings raise the possibility that miR-302 modulation could be harnessed for regenerative applications. Expanding or maintaining pancreatic progenitor pools *in vitro*, or enhancing islet regeneration in diabetes, may benefit from targeted manipulation of miR-302 levels. Given the evolutionary conservation of miR-302 (Chen et al., 2015; Gao et al., 2015), investigating its functions in mouse pancreatic development could shed light on human congenital pancreatic defects and inform therapeutic strategies. Together, our study opens new avenues to understand the molecular logic of pancreatic development and sets the stage for investigating how miRNA pathways intersect with classical morphogen signaling and transcriptional control to regulate organogenesis.

## MATERIALS AND METHODS
### Animals
All animal work procedures were approved by the Baylor College of Medicine Institutional Animal Care and Use Committee (AN-7033) and performed in accordance with regulations and established guidelines. For adult mouse genotyping, 1-2 mm ear tissues were clipped and lysed using 75 µl of a 25 mM NaOH and 0.2 mM EDTA buffer at 98°C for 1 h and neutralized with 75 µl of 40 mM Tris-Cl, pH 5.5. For embryo genotyping, yolk sac tissue was digested overnight in lysis buffer [50 mM Tris-HCl (pH 8.0), 10 mM EDTA, 100 mM NaCl, 0.1% SDS, and 10 mg/ml proteinase K]. Cell debris was removed by spinning at 13,000 rpm for 10 min, and the supernatant was mixed with an equal amount of isopropanol to precipitate DNA at −30°C for 2 h. DNA was pelleted by a 30-min centrifugation at 13,000 rpm, and then washed with 70% ethanol, dried, and resuspended in TE buffer. Genotyping PCR for all alleles was performed via a 64°C-58°C touchdown PCR program with a total of 40 cycles (annealing at 64°C, 62°C, and 60°C for five cycles each, followed by 58°C for 25 cycles). All PCR primers and expected product sizes are in Table S8.

For lineage tracing, *mir-302-eGFP* reporter mice (Keuls et al., 2023; Parchem et al., 2015), in which the eGFP coding sequence replaced the *mir-302* locus, were used to visualize *mir-302* expression. Transgenic *Pdx1-Cre* (Gu et al., 2002) and *Foxa2*^{mcm} CreER knock-in (Park et al., 2008) alleles were to label pancreatic progenitors and endoderm. *Pdx1-Cre* (Gu et al., 2002), *mir-302*^{GFP/+} or *Foxa2*^{mcm} (Park et al., 2008), *mir-302*^{GFP/+} (Keuls et al., 2023) males were crossed with *Rosa26*^{LSL-tdTomato/LSL-tdTomato} (Madisen et al., 2010) females, and noon of the day discovering the vaginal plug was designated as E0.5. To label endoderm using *Foxa2*^{mcm}, 0.05 mg of tamoxifen per gram of body weight was injected intraperitoneally into pregnant dam at E6.5 (Park et al., 2008). To knock out *mir-302*, *Pdx1-Cre*, *mir-302*^{+/−} males were crossed with *mir-302*^{+/−}, *Rosa26*^{LSL-tdTomato/LSL-tdTomato} females. For BrdU incorporation assay, 100 mg of BrdU (B5002, Sigma-Aldrich) per kilogram of body weight was injected intraperitoneally into pregnant dam 30 min before embryo dissection.

### Tissue processing and staining
Mouse embryos were dissected in phosphate-buffered saline (PBS) at pH 7.4 at noon on designated day and fixed overnight at 4°C in 3.7% formaldehyde (BP531-500, Fisher BioReagents, diluted in PBS). Following fixation, embryos were washed with PBS and methanol, stored in methanol at −30°C until use. For cryosectioning, embryos were rehydrated with PBS and equilibrated in 10%, 20%, and 30% w/v sucrose in PBS, and lastly in 100% OCT Compound (Thermo Fisher Scientific, 23730571). Embryos were embedded in OCT, flash-frozen in a dry ice/100% ethanol bath and stored at −80°C. 10-µm serial sections were collected for precise quantification across different area in the pancreas. Slides were stored at −80°C until staining. For E17.5 embryos, the liver, pancreas, spleen, and intestine were dissected together from the embryos, and embedded and sectioned together coronally.

For immunofluorescence, slides were brought to room temperature (RT), washed three times with 0.1% Triton–X in PBS (PT) for 5 min each, and blocked (5% goat serum, 1% BSA in PBS) at RT for 1 h. The diluted primary antibody solution was centrifuged at 15,000× *g* for 10 min at 4°C before use and incubated overnight at 4°C. After incubation, slides were washed five times with PT for 5 min, then incubated with fluorescently conjugated secondary antibodies and DAPI (1 µg/ml, Sigma-Aldrich) for 2 h at RT. Slides were washed five times with PT and mounted in Fluoromount-G (OB10001, Southern Biotech). Images were taken on an inverted fluorescent microscope (LSM 980 confocal microscope; Zeiss, Thornwood, NY, USA). Primary antibodies used in this study and their dilutions were: GFP (ab13970, Abcam, 1:500), Pdx1 (F6A11, DSHB, 1:400), insulin (sc-8033, Santa Cruz Biotechnology, 1:500), glucagon (sc-514562, Santa Cruz Biotechnology, 1:500), E-cadherin (3196, Cell Signaling Technology, 1:200), mucin1 (MA5-11202, Invitrogen, 1:500), BrdU (ab6326, Abcam, 1:500), phospho-histone H3 (9701, Cell Signaling Technology, 1:500). Antigen retrieval was done in Buffer A (62706-10, EMS) only when staining for BrdU and pHH3.

Biology Open

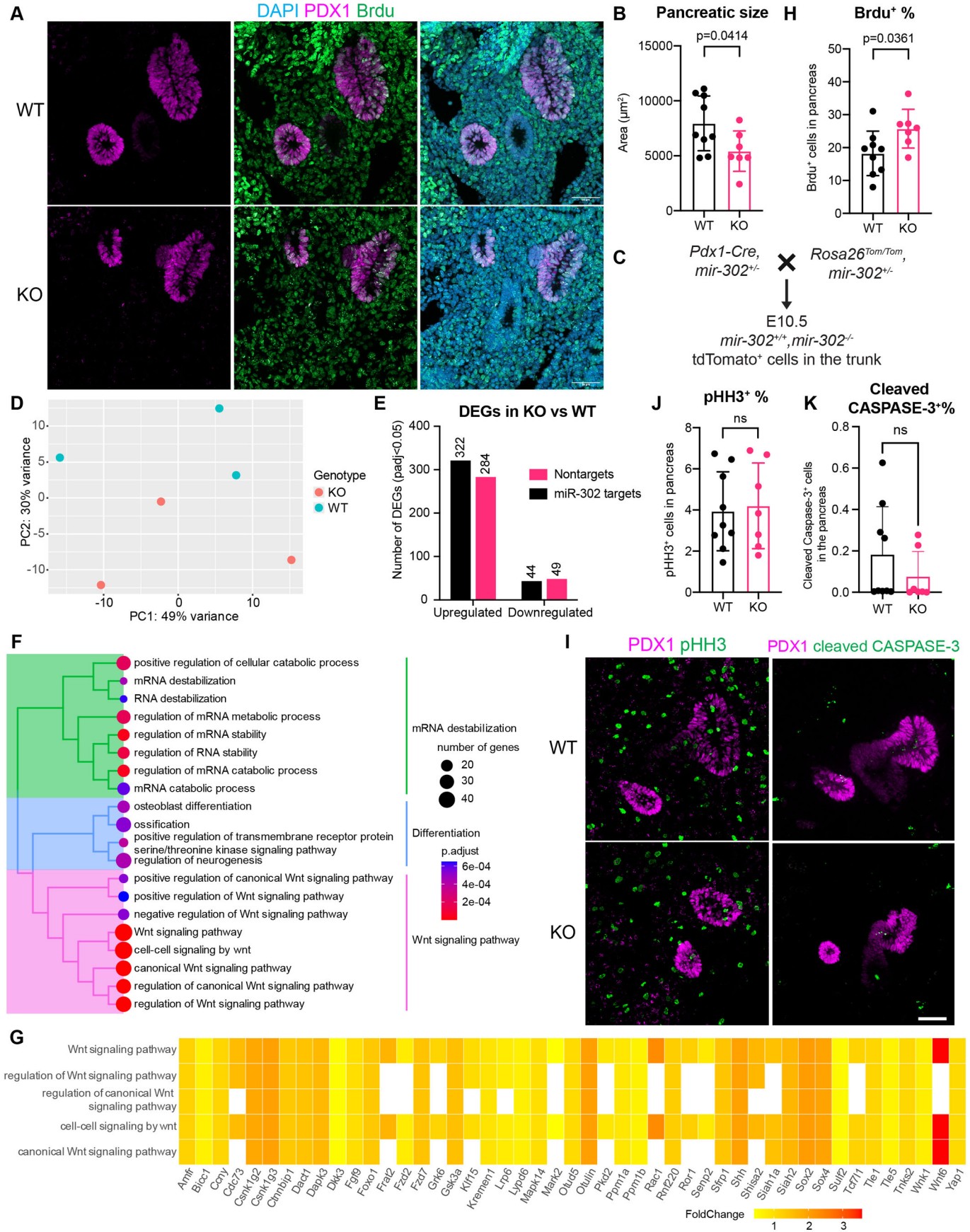

**Fig. 4.** See next page for legend.

**Fig. 4. *mir-302* controls pancreatic progenitor pool and size.** (A) IF of PDX1 (magenta) and BrdU (green) after a 30-min pulse at E10.5. Scale bars: 50 µm. (B) Quantification of PDX1+ area in A. (C) Schematic of lineage tracing strategy to isolate *mir-302* WT and KO MPCs at E10.5 to compare transcriptome. (D) PCA plot of WT and KO samples based on transcriptomic data. (E) Number of DEGs in KO versus WT. DEGs were obtained using DESeq2 with an adjusted *P*-value <0.05. (F) GO enrichment of upregulated DEGs. (G) Heatmap showing the expression fold change (KO versus WT) of genes related with Wnt signaling pathway. (H) Quantification of BrdU+ pancreatic cells in A. (I) IF of phospho-histone H3 (green) and PDX1 (magenta) (left), and cleaved CASPASE-3 (green) and PDX1 (magenta) (right) in KO and WT pancreas at E10.5. Scale bar: 50 µm. (J) Quantification of pHH3+ cells in pancreatic cells in I. (K) Quantification of cleaved CASPASE-3+ cells in pancreatic cells in I. *P*-value in B,H,J,K by Student's *t*-test. Each data point represents an embryo quantified with at least two sagittal sections containing pancreatic cells.

## Image quantification and statistical analysis

Sections throughout the whole gastrointestinal tract were stained and quantified. Images were quantified using Fiji (Schindelin et al., 2012). Background intensity was subtracted from each imaging via threshold function. The pancreatic area was determined by outlining the PDX1+ area, and INSULIN, GLUCAGON, BrdU, MUC1+ area percentage within the PDX1+ pancreas was used in quantification. Each dot represents the average quantification in an embryo. At least two sections containing pancreatic cells were quantified at E10.5, and at least 12 sections across the pancreas per embryo was quantified at E17.5.

All quantification data were analyzed and plotted with GraphPad Prism and are expressed as mean±standard deviation (SD). Differences between two groups were evaluated with an unpaired Student's *t*-test. A *P*-value less than 0.05 was considered statistically significant. ****$P<0.0001$, ***$P<0.001$, **$P<0.01$, *$P<0.05$, ns=not significant.

## Cell sorting

To sort cells from embryos, whole embryo from *Foxa2^mcm* litters and the trunk region for *Pdx1-Cre* litters were dissociated with 100 µl of warm 0.25% trypsin-EDTA at 37°C for 5 min with gentle disturbance (500 rpm) in thermomixer (Eppendorf). After 5 min, tissue was dissociated by pipetting up and down six times and another 1 min of incubation in thermomixer. The enzymatic dissociation was stopped by adding 100 µl of FBS, and 1 ml of PBS. After a centrifugation at 400 *g* for 4 min at 4°C, supernatant was removed, and cell pellet was washed with 1 ml of PBS again. Finally, cells were suspended in FACS buffer (1%BSA, 1 mM EDTA, 10 mM HEPES in PBS) and filtered through cell strainer cap test tube (352235, Falcon). Cells were stained with DAPI and sorted on a FACSAriaII (BD Biosciences) with a 100-µm nozzle into a 1.5 ml tube containing 500ul of Trizol LS (Thermo Fisher Scientific, 10296028).

## RNA extraction and library preparation

Total RNA was purified using the miRNeasy micro kit (217084, Qiagen) according to the manufacturer's instructions. The RNA quality was determined on the high-sensitivity RNA screentape with the Agilent 2200 TapeStation System according to the manufacturer's instructions (Agilent Technologies). Small RNA libraries were generated with the NEXTflex Small RNA-Seq kit v3 (NOVA5132, Bioo Scientific). After PCR amplification, small RNA libraries were purified via TBE-PAGE (4565014, Bio-Rad). For mRNA library prep, cDNA was generated via the SMART-seq v4 ultra-low input RNA kit for sequencing (634889, Takara) and then 1 ng of cDNA was used to make library by the Nextera XT DNA library kit (FC-131-1024, Illumina). All the RNA samples and libraries were quantified using DeNovix assay kits, and the quality was determined on high-sensitivity screen tape with the Agilent 2200 TapeStation System. Small RNA libraries were sequenced on Nextseq 2000 platform (Illumina) in single-end 50 bp, and mRNA libraries were sequenced on Nextseq 550 (Illumina) paired-end 150 bp.

## Sequencing data analysis

### miRNA

The FASTQ files were downloaded from BaseSpace Sequence Hub (Illumina). The quality control, alignment, and annotation were done following COMPSRA pipeline (Li et al., 2020) using the following parameters: -ref mm10 -qc -ra TGGAATTCTCGGGTGCCAAGG -rb 4 -rh 20 -rt 20 -rr 20 -rlh 8,17 -aln -mt star -ann -ac 1, and then mapped reads were quantified with -fun -fm -fms 1-12 -fdclass 1,2 -fdann -pro COMPSRA_MERGE. Raw read counts for individual miRNAs with the same seed sequence (Kozomara et al., 2019) were summed to calculate raw counts for each miRNA family. The DESeq2 package (Love et al., 2014) was used to normalize read counts. K-means clustering was done using miRNAs with normalized count >100 in at least three samples. Normalized read counts were obtained using DESeq2 and scaled with Z-score normalization for K-means clustering.

To identify the most representative miRNAs in each expression cluster, the Euclidean distance between each miRNA's scaled expression and the centroid of its assigned cluster was calculated. The cluster centroids were defined as the mean expression vectors across all miRNAs within each cluster. For each cluster, miRNAs were ranked by increasing distance to the corresponding centroid. The top five miRNAs per cluster were selected as representative candidates reflecting the core expression trajectory of that group.

### mRNA

The FASTQ files were downloaded using the BaseSpace Sequence Hub. Low-quality sequences and bases were removed before downstream analysis. Sample quality and read numbers were assessed using FastQC (Andrews). The STAR package (Dobin et al., 2013) was used to align reads to GRCm39 using default parameters. The GenomicAlignments package (Lawrence et al., 2013) was used to quantify read counts and the DESeq2 (Love et al., 2014) was used to determine DE mRNAs. GO enrichment analysis was done via clusterProfiler (Wu et al., 2021) with significantly upregulated genes (*P*adj<0.05). Bioinformatic analysis was visualized using ggplot2 (Wickham, 2016) and GraphPad Prism software.

## Acknowledgements

We thank the Cytometry and Cell Sorting Core at Baylor College of Medicine for help with FACS. All schematics were created using Biorender illustrations.

## Competing interests

The authors declare no competing or financial interests.

## Author contributions

Data curation: Z.Z.Y.; Formal analysis: Z.Z.Y., C.G.S.; Funding acquisition: R.J.P.; Investigation: Z.Z.Y., C.G.S.; Methodology: Z.Z.Y., C.G.S.; Resources: R.J.P.; Software: Z.Z.Y.; Supervision: R.J.P.; Validation: Z.Z.Y.; Visualization: Z.Z.Y.; Writing – original draft: Z.Z.Y.; Writing – review & editing: Z.Z.Y., R.J.P.

## Funding

This work was supported by the Cancer Prevention and Research Institute of Texas Scholar in Cancer Research (CPRIT RR150106 to R.J.P.), the Andrew McDonough B+ Foundation (R.J.P.), and the V Foundation for Cancer Research (V2017-017 to R.J.P.). The Parchem lab was also supported by the National Institute of Child Health and Human Development (R01HD099252 and R01HD098131). Open Access funding provided by NIH. Deposited in PMC for immediate release.

## Data and resource availability

Sequencing data generated in this study have been deposited in the Gene Expression Omnibus (GEO) under the accession numbers GSE297840 and GSE297841. Additional information is available from the corresponding author upon reasonable request. All relevant data and details of resources can be found within the article and its supplementary information.

## Peer review history

The peer review history is available online at https://journals.biologists.com/bio/lookup/doi/10.1242/bio.062353.reviewer-comments.pdf

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
