## [Peer Review File · Biology Open]

miR-302 regulates pancreatic progenitor pool and pancreatic size

Ziyue Yang, Caroline Snider, Ronald Parchem

DOI: 10.1242/bio.062353

Editor: Tristan Rodríguez

Review timeline

Original submission: 3 November 2025

Editorial decision: 7 November 2025

First revision received: 7 December 2025

Accepted: 8 December 2025

Original submission

First decision letter

MS ID#: bio.062353

MS Title: MiR-302 regulates pancreatic progenitor pool and pancreatic size

Authors: Ziyue Yang, Caroline Snider, Ronald Parchem

Article Type: Research Article

Dear Dr Parchem,

I have now reached a decision on the above manuscript.

The reviewer reports are shown at the bottom of this email.

As you will see, the reviewers gave favourable reports, but raised some critical points that will require amendments to your manuscript. I hope that you will be able to carry these out, because we would like to be able to accept your paper.

At this stage, we also ask you to ensure your manuscript complies with our formatting guidelines - please see our manuscript preparation guidelines for details. Provided you are able to fully address the referees' comments, we are positive about publication of your paper (we accept over 95% of revision submissions) and therefore hope you won't mind any extra work involved in reformatting your manuscript at this point.

Reviewer 1

Comments for the author

In this manuscript, the authors profile miRNA expression in multipotent pancreatic progenitor cells (MPCs), identifying miR-302 as a key regulator of early pancreatic development. The authors present compelling data showing that miR-302 loss results in a smaller pancreas without altering cell fate, and they mechanistically link this to cell cycle progression and Wnt signaling. The study is

well-written, the figures are generally clear, and the conclusions are mostly supported by the data. However, some key claims require additional experimental support and clarifications to strengthen the manuscript.

Major Comments:

1. Evidence for S-phase Extension in MPCs.

The claim that miR-302 loss prolongs S-phase is central to the manuscript's mechanism but relies solely on a 30-minute BrdU pulse (Figure 4H). While the increased BrdU incorporation is consistent with a longer S-phase, this experiment alone cannot rule out alternative explanations, such as a concurrent, transient increase in proliferation followed by apoptosis.

To robustly support this conclusion, a BrdU or EdU pulse-chase experiment is recommended. For instance, administering a pulse of BrdU or EdU followed by a chase period (eg a few hours) would allow the authors to track the retention of the label in PDX1+ MPCs. A higher retention of EdU in PDX1+ MPCs in the mutants after a chase would provide direct evidence for a delayed cell cycle progression and extended S-phase duration.

To address the potential role of apoptosis in the reduced organ size, staining for cleaved caspase-3 at E10.5 and E17.5 would be informative. This would help clarify if cell death contributes to the smaller MPC pool and final pancreas size.

2. Clarification of Mouse Models and Genetic Lineage Tracing.

The description of the mouse models and genetic strategies needs to be more precise and detailed.

Mouse Nomenclature: Please use standard genetic nomenclature throughout and clearly state which alleles are knock-ins, transgenics, or knockouts in the methods.

Lineage Tracing Experiment (Section: 'miR-302 is expressed during the primary transition'): The logic and results of this experiment are currently difficult to follow. The relationship between the miR-302 reporter, the Pdx1-Cre allele, and the resulting GFP expression patterns needs a more thorough explanation. A clearer description of the genetic cross and the expected versus observed outcomes is essential for the reader to interpret these data.

Minor Comment:

3. miRNA Clustering Analysis (Figure 1D).

The authors note that no dynamically expressed miRNAs were found in a pattern opposite to cluster 3 (i.e., down-then-up). It would be helpful to briefly comment on whether this was a genuine biological finding or a limitation of the statistical thresholds used in the clustering analysis.

In summary, this study presents a valuable resource and an interesting finding. Addressing the points above, particularly by providing stronger evidence for the S-phase extension and clarifying the genetic tools, would significantly strengthen the manuscript and its impact.

Reviewer 2

Comments for the author

This manuscript examines the role of miR-302 in early pancreatic development, demonstrating that miR-302 is enriched during the primary transition (E8.5-E9.5) and contributes to the expansion of the multipotent pancreatic progenitor pool, thereby influencing overall pancreatic size. The authors use appropriate genetic models, lineage tracing, proliferation assays, and transcriptomic analysis. The experimental approach is sound, with proper controls and suitable statistical

methods. The number of biological replicates appears adequate, and the methods are well described with sufficient detail to support reproducibility.

The data broadly support the conclusions. While mechanistic specificity is correlative, the authors acknowledge this in the Discussion and note that multiple transcriptional regulators and signalling pathways influence MPC expansion, and that what miR-302 could be regulating is speculative. That said, the study is well situated within the pancreatic development literature, and citations are comprehensive.

The only other recommendation I would make is to ensure that there is an explicit statement about the deposition of the RNA-Seq data.

Reviewer's Responses to Questions

Experimental quality

Does each figure have the proper controls?

If 'No', please indicate reasons in Comments for Author box below.

Reviewer #1:

- Yes

Reviewer #2:

- Yes
-

Were the data analyzed using appropriate statistical tests?

If 'No', please indicate reasons in Comments for Author box below.

Reviewer #1:

- Yes

Reviewer #2:

- Yes
-

Reproducibility

Were experiments performed using adequate number of biological replicates?

If 'No', please indicate reasons in Comments for Author box below.

Reviewer #1:

- Yes

Reviewer #2:

- Yes
-

Does the methods section provide sufficient detail to permit reproducibility?

If 'No', please indicate reasons in Comments for Author box below.

Reviewer #1:

- Yes

Reviewer #2:

- Yes
-

Completeness

Are the manuscript's conclusions supported by the data?

If 'No', please indicate reasons in Comments for Author box below.

Reviewer #1:

- No

Reviewer #2:

- Yes

Scholarship

Do the authors cite and discuss the merits of data that would argue for and against their conclusion?

If 'No', please indicate reasons in Comments for Author box below.

Reviewer #1:

- Yes

Reviewer #2:

- Yes

Does the manuscript title & abstract accurately reflect the contents of the manuscript, without hyperbole?

If 'No', please indicate reasons in Comments for Author box below.

Reviewer #1:

- Yes

Reviewer #2:

- Yes

First revisionAuthor response to reviewers' comments

We are very grateful to all the reviewers for their constructive and thoughtful feedback. We think that our modifications to the manuscript in response to the reviewers' comments have significantly improved the paper. Below the reviewer comments are in grey, our responses are in black, and changes in the manuscript are highlighted in purple. We hope that with these changes the revised manuscript will now be acceptable for publication.

Reviewer 1: In this manuscript, the authors profile miRNA expression in multipotent pancreatic progenitor cells (MPCs), identifying miR-302 as a key regulator of early pancreatic development. The authors present compelling data showing that miR-302 loss results in a smaller pancreas without altering cell fate, and they mechanistically link this to cell cycle progression and Wnt signaling. The study is well-written, the figures are generally clear, and the conclusions are mostly supported by the data. However, some key claims require additional experimental support and clarifications to strengthen the manuscript.

Major Comments:

1. Evidence for S-phase Extension in MPCs.

The claim that miR-302 loss prolongs S-phase is central to the manuscript's mechanism but relies solely on a 30-minute BrdU pulse (Figure 4H). While the increased BrdU incorporation is consistent with a longer S-phase, this experiment alone cannot rule out alternative explanations, such as a concurrent, transient increase in proliferation followed by apoptosis.

To robustly support this conclusion, a BrdU or EdU pulse-chase experiment is recommended. For instance, administering a pulse of BrdU or EdU followed by a chase period (eg a few hours) would allow the authors to track the retention of the label in PDX1+ MPCs. A higher retention of EdU in PDX1+ MPCs in the mutants after a chase would provide direct evidence for a delayed cell cycle progression and extended S-phase duration.

The reviewer raised an excellent point. To directly assess S-phase progression, we set up mouse crosses and administered a pulse of BrdU followed by a chase period of 2 hours before harvesting embryos at E10.5 (Reviewers' Figure 1). Despite current limited mouse colony availability, we were able to collect four KO embryos. The data show a clear trend toward increased BrdU retention in PDX1+ MPCs in the miR-302 KO embryos compared to controls, consistent with a delayed S-phase progression. These results support the conclusion drawn from the initial 30-minute BrdU pulse assay and strengthen our interpretation that miR-302 loss prolongs S-phase in MPCs.

Reviewer's Figure 1. IF of PDX1 (magenta) and BrdU (green) after a 2-hour pulse at E10.5 and quantification. Scale bar=50 μ m.

To address the potential role of apoptosis in the reduced organ size, staining for cleaved caspase-3 at E10.5 and E17.5 would be informative. This would help clarify if cell death contributes to the smaller MPC pool and final pancreas size.

To compare the apoptosis in the pancreas, we analyzed the cleaved caspase-3 at E10.5. At E10.5, there were few cells showed positive staining, and we observed a trend toward reduced apoptosis in the KO compared to WT, although this difference was not statistically significant (Figure 4I, 4K).

There was no significant changes in apoptosis either (Figure 4I, 4K).

Together, these results indicate that apoptosis does not appear to be elevated in miR-302 KO embryos and therefore is unlikely to account for the reduced MPC pool or smaller pancreas size. This supports our interpretation that prolonged S-phase in MPCs contributes to decreased progenitor expansion and reduced organ size.

2. Clarification of Mouse Models and Genetic Lineage Tracing.

The description of the mouse models and genetic strategies needs to be more precise and detailed.

Mouse Nomenclature: Please use standard genetic nomenclature throughout and clearly state which alleles are knock-ins, transgenics, or knockouts in the methods.

Lineage Tracing Experiment (Section: 'miR-302 is expressed during the primary transition'): The logic and results of this experiment are currently difficult to follow. The relationship between the miR-302 reporter, the Pdx1-Cre allele, and the resulting GFP expression patterns needs a more thorough explanation. A clearer description of the genetic cross and the expected versus observed outcomes is essential for the reader to interpret these data.

We apologize for the lack of clarity and have revised the Methods and Results sections to more precisely describe the mouse models and genetic strategies used.

- *Pdx1-Cre* (Gu et al., 2002): transgenic Cre driver
- *mir-302-eGFP* (Parchem et al., 2015): *mir-302* knock-in reporter mice, in which the coding sequence for eGFP replaced the *mir-302* locus. GFP allele also functions as knockout. *mir-302*^{+/+} (wildtype), *mir-302*^{GFP/+} (heterozygous *mir-302* mutant) and *mir-302*^{GFP/GFP} (homozygous *mir-302* mutant)
- *mir-302*^{+/-} (Keuls et al., 2023): heterozygous *mir-302* mutant
- *Rosa26*^{LSL-tdTomato} (Madisen et al., 2010): Ai9, Cre-dependent tdTomato knock-in reporter
- *Foxa2*^{mcm} (Park et al., 2008): Cre^{ER} knock-in allele

For the lineage tracing experiment, *mir-302-eGFP* knock-in allele was used to directly report *mir-302* expression and is not regulated by *Pdx1-Cre*, which were to label pancreatic progenitors. When crossing *Pdx1-Cre*, *mir-302*^{GFP/+} with Ai9 homozygous, all embryos will inherit the Ai9 reporter, and those carrying both *Pdx1-Cre* and *mir-302*^{GFP} allele will show *mir-302* expression (GFP+) in pancreatic progenitors (tdTomato+).

We also updated nomenclature throughout the manuscript.

To visualize *mir-302* expression within pancreatic lineages, we employed a knock-in *mir-302-eGFP* reporter and simultaneously used *Pdx1-Cre* (Parchem et al., 2015) driving *Rosa26*^{LSL-tdTomato} (Madisen et al., 2010) to label pancreatic progenitors (Figure 2C).

For lineage tracing, *mir-302-eGFP* reporter mice (Keuls et al., 2023; Parchem et al., 2015), in which the eGFP coding sequence replaced the *mir-302* locus were used to visualize *mir-302* expression. Transgenic *Pdx1-Cre* (Gu et al., 2002) and *Foxa2*^{mcm} CreER knock-in (Park et al., 2008) alleles were to label pancreatic progenitors and endoderm lineages.

Minor Comment:

3. miRNA Clustering Analysis (Figure 1D).

The authors note that no dynamically expressed miRNAs were found in a pattern opposite to cluster 3 (i.e., down-then-up). It would be helpful to briefly comment on whether this was a genuine biological finding or a limitation of the statistical thresholds used in the clustering analysis.

Thank you for pointing this out. Upon re-examining the clustering results, we found that miRNAs with a down-then-up pattern do exist, but were not highlighted previously due to our clustering thresholds and the parameters. Importantly, these miRNAs showed relatively modest amplitude changes, which may explain why they were not emphasized initially. We have clarified this in the manuscript and expanded the description to note that such trajectories are present.

This analysis identified three major clusters of miRNAs during early pancreatic development from E9.5 to E11.5, representing the dominant patterns of: (1) upregulated, (2) downregulated, and (3) dynamically expressed.

In summary, this study presents a valuable resource and an interesting finding. Addressing the points above, particularly by providing stronger evidence for the S-phase extension and clarifying the genetic tools, would significantly strengthen the manuscript and its impact.

Reviewer 2: This manuscript examines the role of miR-302 in early pancreatic development, demonstrating that miR-302 is enriched during the primary transition (E8.5-E9.5) and contributes to the expansion of the multipotent pancreatic progenitor pool, thereby influencing overall pancreatic size. The authors use appropriate genetic models, lineage tracing, proliferation assays, and transcriptomic analysis. The experimental approach is sound, with proper controls and suitable statistical methods. The number of biological replicates appears adequate, and the methods are well described with sufficient detail to support reproducibility.

The data broadly support the conclusions. While mechanistic specificity is correlative, the authors acknowledge this in the Discussion and note that multiple transcriptional regulators and signalling pathways influence MPC expansion, and that what miR-302 could be regulating is speculative. That said, the study is well situated within the pancreatic development literature, and citations are comprehensive.

The only other recommendation I would make is to ensure that there is an explicit statement about the deposition of the RNA-Seq data.

We thank the reviewer for the positive evaluation of our study. We included a section of Data Availability in our manuscript which listed the GEO number of RNA-seq data.

Data Availability

Sequencing data generated in this study have been deposited in the Gene Expression Omnibus (GEO) under the accession numbers GSE297840 and GSE297841. Additional information is available from the corresponding author upon reasonable request.

Reference:

- Gu, G., Dubauskaite, J. and Melton, D. A. (2002). Direct evidence for the pancreatic lineage: NGN3+ cells are islet progenitors and are distinct from duct progenitors. *Development* **129**, 2447-2457.
- Keuls, R. A., Oh, Y. S., Patel, I. and Parchem, R. J. (2023). Post-transcriptional regulation in cranial neural crest cells expands developmental potential. *Proc Natl Acad Sci USA* **120**, e2212578120.
- Madisen, L., Zwingman, T. A., Sunkin, S. M., Oh, S. W., Zariwala, H. A., Gu, H., Ng, L. L., Palmiter, R. D., Hawrylycz, M. J., Jones, A. R., et al. (2010). A robust and high-throughput Cre reporting and characterization system for the whole mouse brain. *Nat. Neurosci.* **13**, 133-140.
- Parchem, R. J., Moore, N., Fish, J. L., Parchem, J. G., Braga, T. T., Shenoy, A., Oldham, M. C., Rubenstein, J. L. R., Schneider, R. A. and Blelloch, R. (2015). miR-302 Is Required for Timing of Neural Differentiation, Neural Tube Closure, and Embryonic Viability. *Cell Rep.* **12**, 760-773.
- Park, E. J., Sun, X., Nichol, P., Saijoh, Y., Martin, J. F. and Moon, A. M. (2008). System for tamoxifen-inducible expression of cre-recombinase from the Foxa2 locus in mice. *Dev. Dyn.* **237**, 447-453.

Second decision letter

MS ID#: bio.062353R1

MS Title: MiR-302 regulates pancreatic progenitor pool and pancreatic size

Authors: Ziyue Yang, Caroline Snider, Ronald Parchem

Article Type: Research Article

Dear Dr Parchem,

I am happy to tell you that your manuscript has been accepted for publication in Biology Open, pending our standard publication integrity checks. It was accepted on 8th December 2025.